# SNP rs9364554 Modulates Androgen Receptor Binding and Drug Response in Prostate Cancer

**DOI:** 10.3390/biom15010064

**Published:** 2025-01-04

**Authors:** Yuqian Yan, Lei Shi, Tao Ma, Liguo Wang, Haojie Huang

**Affiliations:** 1Department of Biochemistry and Molecular Biology, Mayo Clinic College of Medicine and Science, Rochester, MN 55905, USA; yan.yuqian@mayo.edu; 2Department of Neurosurgery, Mayo Clinic College of Medicine and Science, Rochester, MN 55905, USA; 3Department of Radiation Oncology, Cancer Center, Zhejiang Provincial People’s Hospital, Affiliated People’s Hospital of Hangzhou Medical College, Hangzhou 310025, China; shileihmu@gmail.com; 4Division of Computational Biology, Department of Quantitative Health Sciences, Mayo Clinic College of Medicine and Science, Rochester, MN 55905, USA; 5Department of Urology, The First Affiliated Hospital, Zhejiang University School of Medicine, Hangzhou 310003, China

**Keywords:** SNP rs9364554, SLC22A3, AR, FOXA1, drug efficacy, prostate cancer

## Abstract

(1) Background: Prostate cancer treatment efficacy is significantly influenced by androgen receptor (AR) signaling pathways. SLC22A3, a membrane transporter, has been linked to SNP rs9364554 risk loci for drug efficacy in prostate cancer. (2) Methods: We examined the location of SNP rs9364554 in the genome and utilized TCGA and other publicly available datasets to analyze the association of this SNP with *SLC22A3* transcription levels. We verified onco-mining findings in prostate cancer cell lines using quantitative PCR and Western blots. Additionally, we employed electrophoretic mobility shift assay (EMSA) to detect the binding affinity of transcription factors to this SNP. The ChIP-Seq was used to analyze the enrichment of H3K27ac on the *SLC22A3* promoter. (3) Results: In this study, we revealed that SNP rs9364554 resides in the *SLC22A3* gene and affects its transcription. The downregulation of SLC22A3 is associated with drug resistance. More importantly, we found that this SNP has different binding affinities with transcription factors, specifically FOXA1 and AR, which significantly affects their regulation of *SLC22A3* transcription. (4) Conclusions: Our findings highlight the potential of using this SNP as a biomarker for predicting chemotherapeutic outcomes and uncover possible mechanisms underlying drug resistance in advanced prostate cancers. More importantly, it provides a clinical foundation for targeting FOXA1 to enhance drug efficacy in prostate cancer patients.

## 1. Introduction

Prostate cancer is a leading cause of cancer-related deaths among men [1] with androgen receptor (AR) signaling playing a pivotal role in its development and progression [2]. The AR functions as a transcription factor (TF) that regulates genes essential for prostate cancer cell proliferation and survival [3]. Androgen deprivation therapy (ADT) is a standard treatment for advanced prostate cancer, aiming to reduce androgen levels and inhibit AR activity [4]. However, despite initial responsiveness, many patients eventually develop castration-resistant prostate cancer (CRPC), or even neuroendocrine prostate cancer (NEPC), which is a more aggressive and treatment-resistant form of the disease [5]. Thus far, the underlying mechanism for drug resistance remains largely unknown.

The ability of AR to bind to its target genes is not solely dependent on its presence but also on the chromatin landscape, which can either facilitate or hinder its access to DNA [6,7]. FOXA1, a member of the Forkhead box (FOX) family, is known for its capacity to bind to compacted chromatin and induce an open chromatin state, facilitating the binding of other TFs [8,9]. Canonically, FOXA1 has been recognized for its role in enhancing AR binding to specific genomic loci, thereby promoting the transcription of AR target genes [10]. However, emerging evidence suggests that FOXA1 can also act as a suppressor of AR binding in certain contexts [11,12].

SNPs are variations at a single nucleotide position in the genome, and they can affect gene function and expression, thereby influencing the pharmacokinetics and pharmacodynamics of therapeutic agents in prostate cancers [13,14]. Among these, SNP rs9364554 has been identified as a variant associated with prostate cancer risk and drug efficacy [15,16,17,18]. This SNP is located on chromosome 6, specifically within the gene SLC22A3, which encodes the organic cation transporter 3 (OCT3) involving in drug-metabolism and transportation [15,19]. SLC22A3 is increasingly being recognized as an important modulator of human disease and drug response due to its involvement in drug absorption and elimination [20]. Mechanistically, SLC22A3 acts as a passive membrane transporter to facilitate drug diffusion [20]. Recent genome-wide association study (GWAS) studies have linked SLC22A3 to SNP rs9364554 risk loci for prostate cancer, colorectal cancer and other diseases [21]. However, whether this SNP affects SLC22A3 gene expression and subsequently undermines drug efficacy is largely unknown.

This study highlights the importance of SNP rs9364554 in modulating FOXA1 binding affinity and its pioneering activity, which leads to differential AR binding and transcriptional outcomes, influencing the drug efficacy. It provides scientific clues in which CRPC or NEPC develop drug resistance through downregulating SLC22A3 transcription. More importantly, considering that CRPC heavily depends on costly chemotherapies, including Taxanes, Enzalutamide, Abiraterone, Radium-233, and Sipuleucel-T [22], understanding the status of SNP rs9364554 appears to be extremely critical for predicting drug efficacy and alleviating unnecessary expenses.

## 2. Materials and Methods

### 2.1. Cell Lines, Cell Culture and Transfection

The PCa cell lines (PC-3, DU145, C4-2, LNCaP, 22RV1 and VCaP) were purchased from ATCC (Manassas, VA, USA). The HEK293T, DU145 and VCaP cell lines were cultured in Dulbecco’s modified Eagle’s medium (DMEM) supplemented with 10% of FBS (Thermo Fisher Scientific, Waltham, MA, USA). PC-3, C4-2, LNCaP, and 22RV1 cell lines were cultured in RPMI 1640 medium supplemented with 10% of FBS. The cells were maintained in a 37 °C humidified incubator supplied with 5% CO_2_.

Lentiviral sgRNA constructs as well as packaging vectors were transfected into HEK293T cells with Lipofectamine 2000 (Thermo Fisher Scientific, Cat# 11668500). Control or gene-specific siRNAs were transfected using Lipofectamine^®^ RNAiMAX (Thermo Fisher Scientific, Cat# 13778) according to the manufacturer’s instruction. Approximately 75~90% transfection efficiencies were routinely achieved. The sequence information for sgRNA and siRNA, and packaging vectors used for lentivirus transfection, are listed in Appendix A.

### 2.2. Cell Proliferation and Clonogenic Survival

For cell proliferation assay, cells were seeded at a density of 1500 cells/well in 96-well plates for 4 h before drug treatment. After 72 h treatment, the cells were incubated with CellTiter 96 AQ one solution (Fisher, Hampton, NH, USA, PRG3580) for 2 h in a 37 °C incubator. The absorbance was read by plate reader at a wavelength of 492 nm. For clonogenic survival assay, it was conducted as previously described [23]. Specifically, C4-2 cells were seeded at a density of 1000 cells/well in 6-well plates. The following day, the cells were treated with DMSO or Enzalutamide/JQ1 for 4 days and then cultured with fresh medium for another 8 days. After 12-day culture, the colonies were fixed with methanol and stained with crystal violet 0.5% (*w*/*v*) for 1 h, which was followed by rinsing with running tap water. The colonies with more than 50 cells were counted.

### 2.3. RNA Extraction, RT-qPCR and ChIP-qPCR

Total RNA was extracted with TRIzol reagent (Invitrogen, Waltham, MA, USA, Cat#15596026) and reversely transcribed into cDNA with a cDNA reverse transcription kit (Thermo Fisher Scientific, Cat# 4368814). The qPCR was performed using iQ SYBR Green Supermix (Bio-Rad, Hercules, CA, USA, Cat# 1708880). The ΔCT was calculated by normalizing the threshold difference of a certain gene with glyceraldehyde-3-phosphate dehydrogenase (GAPDH). The ChIP-qPCR was performed as previously described [24]. Briefly, DNA was routinely pulled down by 5 µg of antibodies or nonspecific IgG for each ChIP reaction. The pull-down products were amplified by quantitative PCR. Primers used for RT-qPCR and ChIP-qPCR are listed in Appendix A.

### 2.4. Nuclear Extraction and Electrophoretic Mobility Shift Assay (EMSA)

Nuclear protein was extracted using NE-PER™ Nuclear and Cytoplasmic Extraction Reagents (Thermo Fisher Scientific, Cat# 78833). EMSA was performed with a LightShift Chemiluminescent EMSA kit (Thermo Fisher Scientific, Cat# 20148) according to a previous publication [25]. Briefly, DNA fragments containing T or C SNP were synthesized and labeled with biotin using a biotin Pierce™ Biotin 3′ End DNA Labeling Kit (Thermo Fisher Scientific, Cat# 89818). The biotin-labelled probes were incubated with nuclear protein extractions (NPEs) for 1 h before loading into 6% of polyacrylamide DNA gel. The antibodies used for EMSA are AR (Santa Cruz, Dallas, TX, sc-13062), AR variant 7 (Precision antibody, Columbia, MD, USA, AG10008), and FOXA1 (Cell Signaling Technology, Danvers, MA, USA, 53528). The probe sequences from the SNP are listed in Appendix A.

### 2.5. DNA Extraction and Sanger Sequencing

The genomic DNA was extracted with a Wizard^®^ Genomic DNA Purification Kit (Promega, Madison, WI, USA, Cat# A1120) from prostate cancer cell lines according to the manufacturer’s instructions, which was followed by PCR to amplify the DNA fragment containing SNP rs9364554. The fragments were purified with a PureLink™ PCR Purification Kit (Invitrogen, Cat# K310002) for Sanger sequencing by the Genewiz company (accessed on 1 January 1999, https://www.genewiz.com/en). The primers for PCR and sequencing are listed below in Appendix A.

### 2.6. Western Blot (WB) Analysis and Antibodies

The WB analysis was performed as described previously [26]. Briefly, cells were harvested and lysed with cell lysis buffer (50 mM Tris-HCl, pH 7.4, 150 mM NaCl, 1% Triton X-100, 1% sodium deoxycholate and 1% protease inhibitor cocktails) on ice for at least 30 min. The cell lysate was sonicated with a Bioruptor Pico sonication system (Diagenode, Denville, NJ, USA) for 30 s on/30 s off program for 15 cycles and then centrifuged for 15 min at 13,000 rpm in 4 °C centrifuge to collect the supernatant. The protein concentration was measured by a Pierce™ BCA Protein Assay Kit (Thermo Fisher Scientific, Cat#23225). The total protein containing 10~50 µg was loaded to 10% SDS-PAGE protein gel and transferred to 0.45 μm nitrocellulose membranes (Thermo Fisher Scientific, Cat#88018). The membrane was blocked with 5% non-fat milk in TBST for 1 h, which was followed by incubation with the appropriate antibody at 4 °C overnight. Primary antibodies were diluted from 1:1000 to 1:2000 with 5% milk in TBST. Secondary antibodies were diluted 1:5000 with 5% milk in TBST.

The primary antibodies used are as follows: SLC22A3 (Abcam, Cambridge, UK, ab124826), β-tubulin (9F3) (Cell Signaling Technology, 2128S), AR (H-280) (Santa Cruz, sc-13062), FOXA1 (Abcam, ab23738), LEF1 (Cell Signaling Technology, 2230S), ERG (BioCare, Hertfordshire, UK, CM421C), IgG (Vector Lab, Newark, CA, USA, I-1000), H3K27ac (Abcam, ab4729) and H3K4me1 (Abcam, ab8895). The secondary antibodies used for WB are Peroxidase AffiniPure™ Goat Anti-Rabbit IgG (H + L) (Jackson ImmunoResearch laboratories, 111-035-003) and AffiniPure™ Goat Anti-Mouse IgG (H + L) (Jackson ImmunoResearch laboratories, West Grove, PA, USA, 111-005-003).

### 2.7. Generation of Graphs and Statistical Analysis

Graphs were generated by using the GraphPad Prism 8 project. All numerical data are presented as mean ± SEM or mean ± SD as required. Differences between groups were compared by unpaired T tests or Chi-square as appropriate. The following symbols were used to denote statistical significance: n.s., not significant, * *p* < 0.05, ** *p* < 0.01, *** *p* < 0.001.

## 3. Results

### 3.1. The SNP rs9364554 Is Highly Associated with Prostate Cancer

Increasing evidence from genome-wide sequencing (GWS) highlights that SNP rs9364554 is highly associated with prostate cancer progression [15,16,17,18]. However, the underlying mechanisms remains unknown. To investigate the relevance of SNP rs9364554 in prostate cancer, we first examined its genomic location and identified that it resides within a topologically associating domain (TAD) encompassing 14 transcribed genes (Appendix A). Specifically, it is located within the SLC22A3 gene alongside two other SLC22 family members (SLC22A1 and SLC22A2). Given that genes within the same TAD physically interact with each other and are regulated by same enhancers, we investigated whether their transcription levels are related to this SNP. Interestingly, the analysis of the TCGA dataset from 335 prostate cancer patients [27] demonstrated that this SNP appears to have no effect on most of these genes except for SLC22A1 and SLC22A3 (Appendix A).

### 3.2. The SLC22A3 Transcription Is Negatively Correlated with Prostate Cancer Progression

To further elucidate the biological meaning of this SNP in prostate cancer, we focused on its regulatory mechanisms affecting SLC22A3 expression for two reasons: (1) among the two genes (SLC22A1 and SLC22A3) influenced by this SNP, SLC22A3 exhibits relatively high transcription expression (Appendix A); (2) functionally, SLC22A3 has been reported to be related to drug efficacy [15,19]. Indeed, the TCGA [27] and other datasets [28] demonstrated that SLC22A3 is downregulated in tumor tissue samples compared with normal compartments (Figure 1A–C), and it is significantly diminished in castration-resistant metastatic prostate cancers [29,30]. Together, all these patients’ data indicate that SLC22A3 is progressively downregulated with prostate cancer malignancy (Figure 1D,E).

To investigate the clinical relevance of SLC22A3 downregulation in prostate cancers, we knocked out SLC22A3 in the prostate cancer C4-2 cell line to assess drug sensitivity to two clinically used anti-prostate cancer drugs, Enzalutamide (AR inhibitor) [31] and JQ1 (BRD4 inhibitor) [32]. Cell proliferation assays demonstrated that although C4-2 cells were sensitive to both Enzalutamide and JQ1, the SLC22A3 knockout dramatically reduced this sensitivity (Figure 1F–H). Further clonogenic survival assays supported this finding, showing that SLC22A3 knockout increased colony numbers after the treatment of either Enzalutamide or JQ1 treatment compared with sgRNA control groups (Figure 1I,J). It is noteworthy that SLC22A3 knockout alone does not affect prostate cancer cell proliferation and survival (Figure 1F–J), suggesting that SLC22A3 is more likely to affect prostate cancer progression via its effects on drug efficacy.

### 3.3. The SNP rs9364554 Affects SLC22A3 Transcription in a Non-Canonical Regulation Pattern

Previous studies revealed that T is a risk allele for SNP rs9364554 [19], prompting us to investigate whether the T allele downregulates SLC22A3 expression. Surprisingly, in normal prostate tissues [27,28], SLC22A3 expression showed no effects on any of the three genotypes (C/C, C/T and T/T), while in prostate cancer tissues [27], patients with a C/T heterozygous background exhibited higher transcription levels of SLC22A3 than those with C/C or T/T homozygous alleles (Figure 2A,B). This finding suggests that this SNP is more likely to be relevant to prostate cancer tissues rather than normal prostate tissues.

To further test the finding in prostate cancer tissues, we sequenced six prostate cancer cell lines and identified their genotypes for this SNP: PC-3 as C/C, DU145 as T/T, and most cell lines (C4-2, LNCaP, 22Rv1 and VCaP) as C/T (Figure 2C). Consistently, SLC22A3 mRNA was expressed at higher levels in cell lines with the heterozygous genotype than those with homozygous genotypes (Figure 2D), which was also reflected at the protein level (Figure 2E). Notably, we found that the SLC22A3 transcription level is positively correlated with AR full-length (FL) expression (Figure 2E,F), indicating that AR FL might enhance SLC22A3 transcription. Indeed, ChIP-Seq analysis has shown that AR specifically binds to the DNA region containing this SNP (Figure 2G).

**Figure 2 biomolecules-15-00064-f002:**
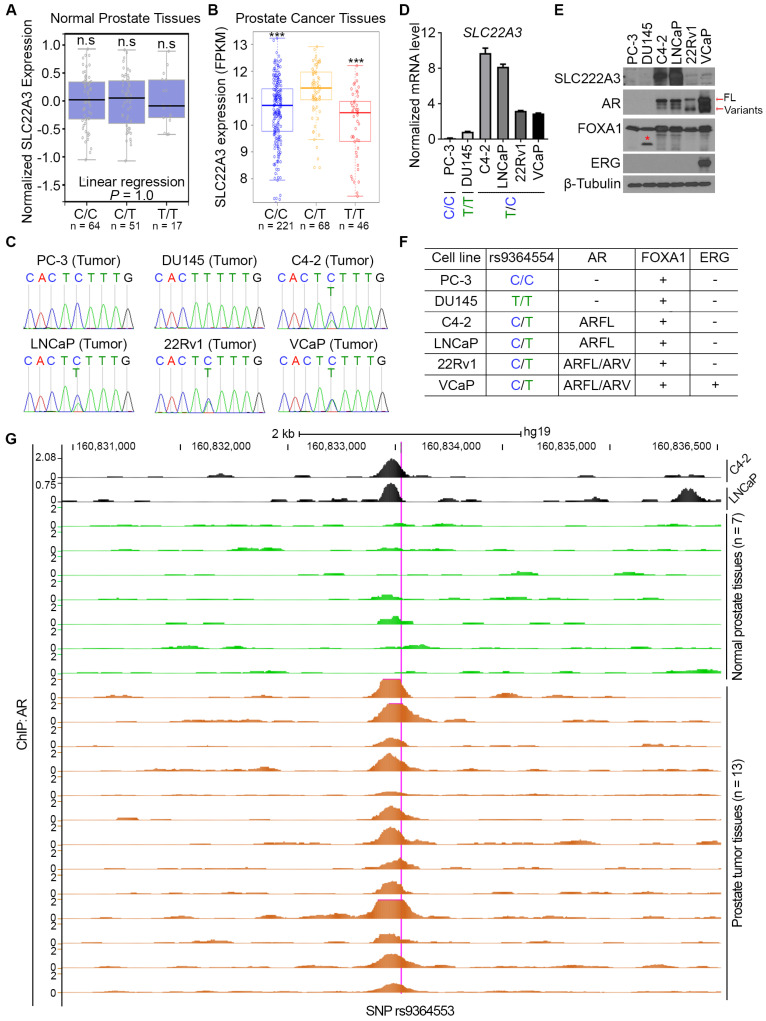
The heterozygous SNP rs9364554 promotes SLC22A3 expression. (**A**,**B**) The SLC22A3 transcription level was compared among different SNP rs9364554 genotypes in patients with normal prostate tissues (**A**) or prostate cancer tissues (**B**). (**C**) The SNP rs9364554 genotyping was performed with Sanger sequencing in six prostate cancer cell lines. (**D**,**E**) The cell line survey was conducted to detect SLC22A3 mRNA (**D**) and protein (**E**) levels. The red “*” indicates a potential FOXA1 mutant protein band. FL: full length. (**F**) A table summarizes genotypes of SNP rs9364554 of six prostate cancer cell lines and their molecular expression status of commonly expressed TFs. (**G**) UCSC tracks from published datasets [33] show patients’ profiles of ChIP-seq signals of AR in this SNP compared with two prostate cancer cell lines (C4-2 and LNCaP). Original images of (**E**) can be found in Appendix A. The following symbols were used to denote statistical significance: *** *p* < 0.001. n.s., not significant.

### 3.4. The SNP rs9364554 Affects TFs Binding with SLC22A3

Previous studies suggest that SNPs affect TF binding sites in promoter regions to subsequently alter the gene expression [34,35]. To investigate whether this is the case for this SNP, we used pseudogene annotation by GENCODE to analyze potential TFs binding at SNP rs9364554. Results from 13 GEO datasets [33,36,37,38,39,40,41,42,43,44,45,46,47], including four prostate cancer cell lines (VCaP, LNCaP, C4-2, PC3), demonstrated that this SNP is within a DNA sequence recognized by several prostate cancer-specific TFs (Figure 3A,B). Among these TFs, the FOXA1 binding motif (TGTTTGC) is quite similar to the sequence where SNP rs9364554 resides (Figure 3C), indicating potential binding. Although the AR binding motif (AGAACA) is not exactly within the sequence containing this SNP, it is adjacent to the FOXA1 binding motif, three nucleotides apart (Figure 3D).

To determine whether the T allele has different binding partners from the C allele, we designed a probe including this SNP (Figure 3E) to perform electrophoretic mobility shift assay (EMSA) with nuclear protein extracts from four prostate cancer cell lines (Figure 3F). The results showed that the C allele formed a protein/DNA complex with nuclear protein extracts from C4-2, VCaP, and DU145 cell lines, but not in AR non-expressing PC-3 cell line (Figure 3F), suggesting that AR might be involved in this complex. Interestingly, we found that the T allele largely diminished the formation of this complex (Figure 3F). Notably, DU145 nuclear protein extracts and the T allele formed a different protein/DNA complex (Figure 3F), which might be associated with a potential truncated FOXA1 protein that was observed in the cell survey (Figure 3E).

To further verify whether this DNA/protein complex contains FOXA1, we added a FOXA1 antibody in the incubation of C/T alleles with VCaP and DU145 nuclear protein extracts. Surprisingly, EMSA results showed that the presence of the FOXA1 antibody did not shift the protein/DNA band (Figure 3G). Consistently, we observed that the T allele impaired the formation of this protein/DNA complex in both VCaP and DU145 cell lines compared with the C allele but formed an evident complex at a lower position with DU145 nuclear protein extracts (Figure 3G).

### 3.5. FOXA1 Undermines AR Binding with SLC22A3 Promoter

Further analysis of the TCGA dataset revealed that FOXA1 mutation significantly downregulated SLC22A3 expression (Figure 4A) [27]. Given that most of FOXA1 mutations are “gain of function”, this indicates that FOXA1 negatively regulates SLC22A3. Indeed, the knockdown of FOXA1 by siRNA elevated SLC22A3 expression, whereas that of AR downregulated SLC22A3 expression in both C4-2 and VCaP cells (Figure 4B). Moreover, the knockdown of ERG did not affect SLC22A3 expression (Figure 4B), which is consistent with TCGA data (Figure 4A). Furthermore, we found that the knockdown of AR decreased H3K27ac enrichment at the promoter of SLC22A3 gene (Figure 4C), supporting the above finding that AR regulates SLC22A3 transcription. Consistently, ChIP-Seq showed that the treatment with an AR inhibitor (Enzalutamide) reduced AR binding and H3K27ac enrichment at the SLC22A3 promoter in C4-2 cells (Figure 4D), while RNA-Seq demonstrated that Enzalutamide diminished SLC22A3 transcription (Figure 4E).

Consistent with the above finding showing that SLC22A3 expressed lower in AR variants (ARV)-expressing cell lines (22Rv1 and VCap) than AR FL-expressing ones (C4-2 and LNCaP) (Figure 2D,E), ChIP-Seq analysis [48] reveals that the expression of ARV undermines AR binding at the SLC22A3 gene (Figure 5A). Meanwhile, although the AR antibody developed a super-shifted band for the protein/DNA complex from the incubation of VCaP nuclear protein extracts with the C allele probe, AR variant 7 antibody failed to develop this shifted band (Figure 5B), suggesting that AR variants have defects on binding to this site. Indeed, the transfection of AR variant 7 (AR-V7) diminished the protein level of SLC22A3 in C4-2 cells (Figure 5C).

Taken together, these results suggest that in primary prostate cancer, SLC22A3 is upregulated in AR FL-expressing cells but diminished in ARV-expressing cells in a background of C allele SNP rs9364554. However, in a background of T allele SNP rs9364554, FOXA1 competes with AR in binding this site and undermines SLC22A3 transcription (Figure 6A). FOXA1 mutations enhance the binding affinity with both C and T alleles, resulting in suppressing AR target gene expression (Figure 6B). In CRPC or NEPC patients, the lack of AR expression impairs SLC22A3 expression (Figure 6C).

## 4. Discussion

Our study provides novel insights into the regulatory mechanisms of the SNP rs9364554 and its association with drug efficacy in prostate cancer through regulating SLC22A3. The analysis from TCGA and other datasets demonstrate that SLC22A3 is downregulated in tumor tissues compared to normal tissues and is further significantly diminished in CRPC. This suggests that lower levels of SLC22A3 are associated with more advanced stages of prostate cancer. Moreover, our studies have first demonstrated that the low expression of SLC22A3 is related to drug resistance, indicating the predictable role of SNP rs9364554 in the drug efficacy of prostate cancer patients.

Contrary to our expectations, the T allele of rs9364554, previously identified as a risk allele [15], did not correlate with lower SLC22A3 expression in normal prostate tissues. However, in prostate cancer tissues, the C/T heterozygous genotype exhibits a higher SLC22A3 transcription level than homozygous genotypes (C/C, T/T), suggesting a complex, non-canonical regulatory mechanism at play in cancerous tissues. Based on the literature and our findings, there are at least two reasons that might contribute to this phenomenon. First, this might be an effect of “molecular heterosis” [49,50], which appears counter-intuitive at a molecular level. Given this SNP is in a regulatory region that recruits TF binding, having two copies (homozygous) would significantly lower gene expression. However, in a heterozygous state, the presence of one normal allele might maintain sufficient TF binding, resulting in higher gene expression. Second, this SNP is in a TAD, which contains an imprinted gene cluster, including the maternally expressed IGF2R, SLC22A2, and SLC22A3 genes and the paternally expressed long non-coding RNA (lncRNA) AIRN [51]. Among these genes, IGF2R and AIRN are reciprocally imprinted [52], which might also lead to the complexity of this regulation.

The SNP rs9364554 resides in a DNA sequence recognized by several prostate cancer-specific TFs, which are altered in different stages of prostate cancer progression or treatment. Hence, the binding of these TFs on this SNP containing DNA is constitutively fluctuated. Especially, AR is expressed in primary prostate cancer, while it loses its expression or develops variants in CRPC or NEPC [53,54]. In this study, we found that either AR loss or ARV expression significantly downregulated SLC22A3 transcription, strongly suggesting SLC22A3 is an AR FL target gene. Considering the importance of SLC22A3 in drug efficacy, this finding highlights the importance of enhancing SLC22A3 expression in combination with other anti-cancer drug in treating CRPC and NEPC patients.

Moreover, our analysis of the TCGA dataset [27] showed that FOXA1 mutations, which are often “gain-of-function” [55], significantly downregulated SLC22A3 expression (Figure 4A). Consistently, FOXA1 knockdown significantly upregulates SLC22A3, suggesting a negative loop of FOXA1 on regulating SLC22A3 transcription. In alignment with previous studies [12], this finding provides another example in which FOXA1 suppresses AR target genes. However, the underlying mechanism is largely unknown. Based on our EMSA results (Figure 3F,G), we observed that DU145 cells developed a strong protein/DNA complex with the T allele. Considering that the DU145 FOXA1 protein has two protein bands (Figure 2E), and the lower band is more likely to be a truncated band, we propose that this complex might be formed with this truncated band. Although the addition of the FOXA1 antibody did not show a shift, that might be because the mutant FOXA1 has lost the epitope for this antibody.

Collectively, our study suggests that the SNP rs9364554 regulates SLC22A3 transcription through a complex interplay of TFs, primarily FOXA1 and AR. Scientifically, this provides an explanation for the resistance of CRPC to most chemotherapies due to an impaired AR signaling pathway. Clinically, it highlights the potential use of this SNP as an indicator to predict drug efficacy in prostate cancer patients. Moreover, it lays the groundwork for further research into the therapeutic potential of targeting FOXA1 to improve drug efficacy in prostate cancer.

## 5. Conclusions

This study provides a novel mechanism by which prostate cancers undermine drug efficacy through the downregulation of SLC22A3 by FOXA1 competition with AR to bind to SNP rs9364554. It highlights the potential use of SNP rs9364554 as a biomarker to predict drug efficacy in precisely treating prostate cancer patients and lays the groundwork for further exploration into the therapeutic potential of targeting FOXA1 to improve drug efficacy in prostate cancer.

## Figures and Tables

**Figure 1 biomolecules-15-00064-f001:**
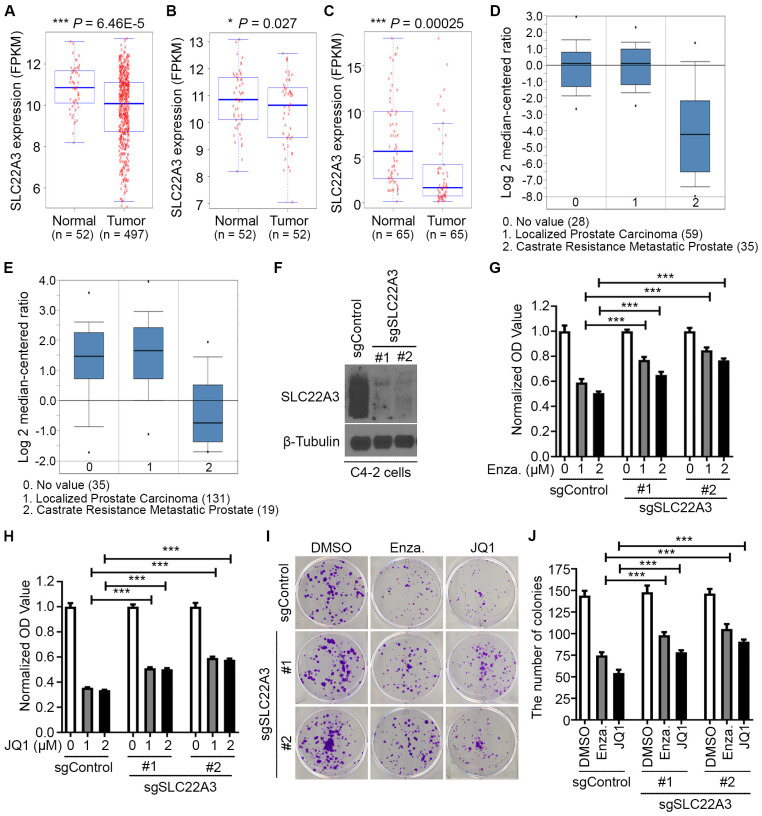
The SLC22A3 expression is highly downregulated in prostate cancer tissues and negatively related to drug efficacy. (**A**,**B**) The SLC22A3 transcription level was compared between prostate normal and cancer tissues from the TCGA patient dataset [27] (**A**) and between paired prostate normal and cancer tissues from the TCGA patient dataset [27] (**B**). (**C**) The SLC22A3 transcription level was compared in patients between paired normal prostate tissues and cancer tissues from a previous study [29]. (**D**,**E**) The SLC22A3 transcription level was compared in patients among non-localized and metastatic prostate tissues from a previous study [30]. (**F**–**H**) The SLC22A3 was knocked out with two sgRNA constructs (#1 and #2) individually in C4-2 cells to detect the SLC22A3 protein level by Western blot analysis in (**F**), and drug treatment in (**G**,**H**). (**I**,**J**) Two sgSLC22A3 knockout cell lines (#1 and #2) were treated with JQ1 (2 µM) and Enzalutamide (2 µM) for clonogenic survival assay. The representative images are shown in (**I**) with quantification data in (**J**). Original images of (**F**) can be found in Appendix A. The following symbols were used to denote statistical significance: * *p* < 0.05, *** *p* < 0.001.

**Figure 3 biomolecules-15-00064-f003:**
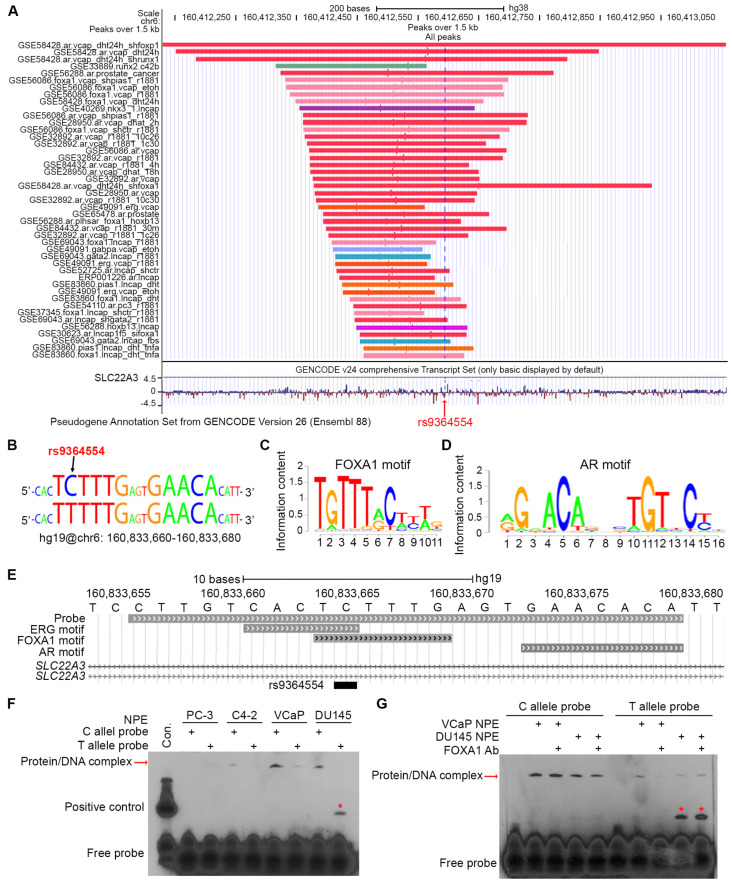
The SNP rs9364554 is manipulated by prostate cancer-specific TFs. (**A**) The online public datasets [33,36,37,38,39,40,41,42,43,44,45,46] reveal that SNP rs9364554 containing a DNA sequence is favorable by multiple prostate cancer-specific TFs (**A**). (**B**–**D**) The DNA sequence containing SNP rs9364554 (**B**) is overlapped with prostate cancer-specific TFs, such as FOXA1 (**C**) and AR (**D**). (**E**–**G**) The probe containing SNP rs9364554 (**E**) was designed to perform EMSA with 1 µg of nuclear protein extracts from four prostate cancer cell lines (PC-3, C4-2, VCaP and DU145) (**F**). The FOXA1 antibody was added to the reactions, which was followed by EMSA (**G**). The red “*” indicates a DNA/protein complex that might contain FOXA1 mutant protein. Original images of (**F,G**) can be found in Appendix A.

**Figure 4 biomolecules-15-00064-f004:**
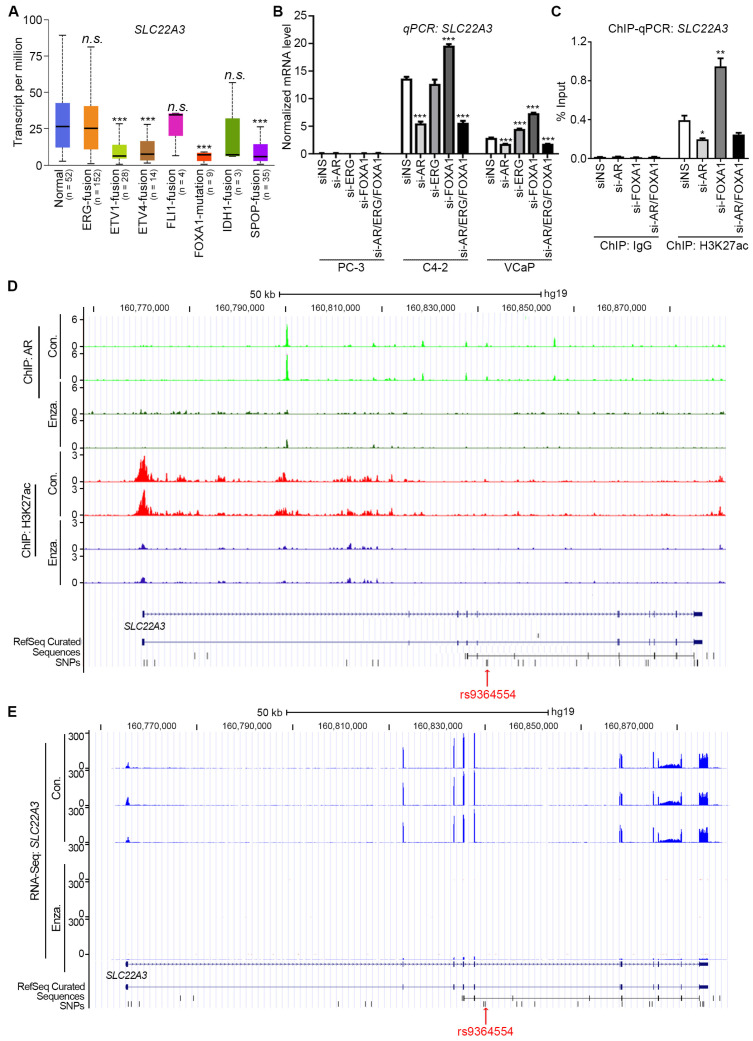
The SLC22A3 transcription is negatively manipulated by FOXA1 while positively regulated by AR. (**A**) The transcription level of SLC22A3 was analyzed according to molecular signature from TCGA datasets with online software (accessed on 21 December 2021, https://ualcan.path.uab.edu/). (**B**,**C**) The C4-2 cells were knocked down with the indicated TFs with siRNA, which was followed by qPCR to detect the mRNA level of SLC22A3 (**B**) and ChIP-qPCR to examine the enrichment of H3K27ac at the SLC22A3 promoter (**C**). (**D**) UCSC tracks from published datasets [48] show profiles of ChIP-seq signals of AR and H3K27ac in C4-2 cells. (**E**) UCSC tracks from published datasets [48] show profiles of RNA-seq signals of SLC22A3 after Enzalutamide treatment in C4-2 cells. The following symbols were used to denote statistical significance: n.s., not significant, * *p* < 0.05, ** *p* < 0.01, *** *p* < 0.001.

**Figure 5 biomolecules-15-00064-f005:**
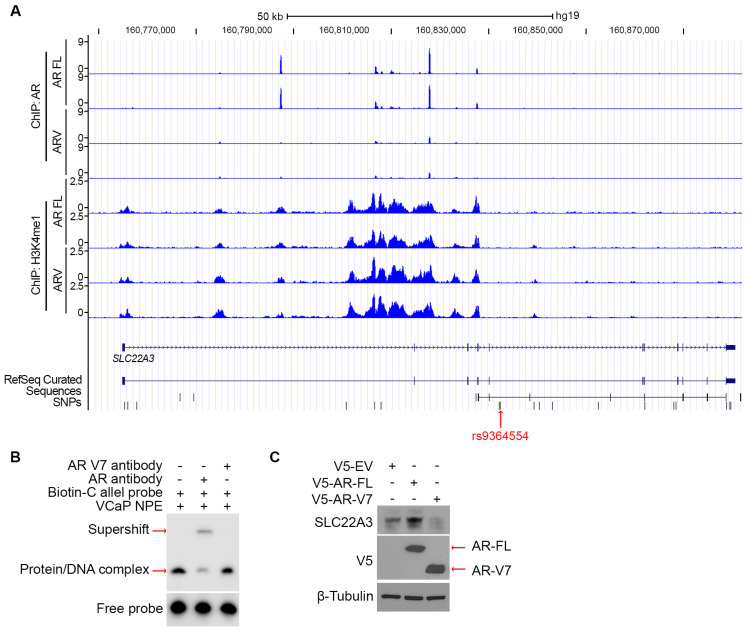
The AR variants undermine SLC22A3 expression. (**A**) UCSC tracks from published datasets [47] show profiles of ChIP-seq signals of AR and H3K4me1 in AR FL and ARV-overexpressing cell lines. (**B**) VCaP nuclear protein extracts were incubated with biotin-labeled C allele, followed by the addition of 0.5 µg of AR or AR variant 7 antibody. The reactions were loaded to 6% of polyacrylamide DNA gel for EMSA assay. (**C**) The C4-2 cells were transfected with AR-FL or -V7 for 48 h and harvested for Western blot to detect the SLC22A3 protein level. Original images of (**B,C**) can be found in Appendix A.

**Figure 6 biomolecules-15-00064-f006:**
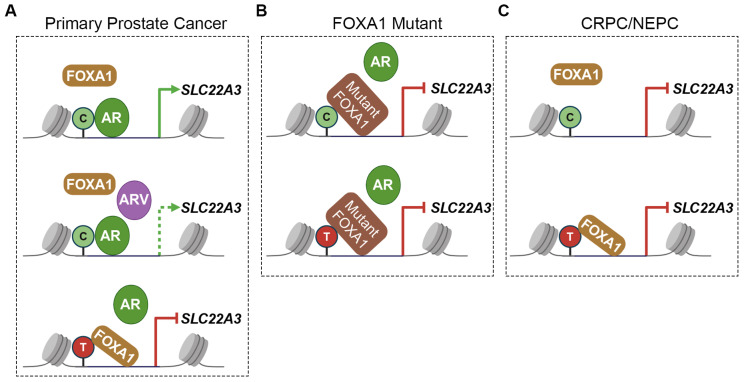
A working model to elucidate the interplay between FOXA1 and AR in regulating SLC22A3 transcription through SNP rs9364554. (**A**) Specifically, in primary prostate cancer, SLC22A3 is upregulated in AR FL-expressing cells but diminished in ARV-expressing cells in a background of C allele SNP rs9364554. However, in a background of T allele SNP rs9364554, FOXA1 competes AR in binding this site and undermines SLC22A3 transcription. (**B**) The mutations of FOXA1 enhance its binding affinity to either the C or T allele, resulting in the suppression of AR target gene expression. (**C**) In CRPC or NEPC patients, the lack of AR expression impairs SLC22A3 expression.

## Data Availability

The original WB and EMSA films are included in Appendix A.

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
