# Peer review of "SNP rs9364554 Modulates Androgen Receptor Binding and Drug Response in Prostate Cancer"

_biomolecules, 2025, doi:10.3390/biom15010064_

Round 1
Reviewer 1 Report
Comments and Suggestions for Authors
The research is highly relevant to one of the cancers with the highest incidence and mortality rates, particularly regarding its progression to CRPC. The manuscript is clear and concise. However, I consider it important to clarify and improve the following points:
In the introduction, the authors state: "This SNP is located on chromosome 6, specifically within the gene SLC22A3, which encodes the organic cation transporter 3 (OCT3), involved in drug metabolism and transportation." Perhaps the authors could delve deeper into the mechanisms underlying the function of SLC22A3 to provide a clearer understanding of its role.
Similarly, the authors mention that SLC22A3 is increasingly being recognized as an important modulator of human disease and drug response; however, they do not explain the underlying mechanisms by which it exerts these effects.
The authors state: "This study highlights the importance of SNP rs9364554 in modulating FOXA1 binding affinity and its pioneering activity, which leads to differential AR binding and transcriptional outcomes, influencing drug efficacy. It provides scientific clues regarding how CRPC or NEPC develop drug resistance through downregulating SLC22A3 transcription. More importantly, the status of SNP rs9364554 can potentially be used as a predictor of drug efficacy for prostate cancer treatment." However, it is important to note that CRPC (castration-resistant prostate cancer) can be approached through various therapeutic strategies, including the use of taxanes, enzalutamide, abiraterone, Radium-233, and sipuleucel-T. The authors may want to address these established treatment modalities to contextualize their findings within the broader therapeutic landscape of CRPC
The authors state: "Lentiviral sgRNA constructs as well as packaging vectors were transfected with Lipofectamine 2000 (Thermo Fisher Scientific, Cat# 11668500)." However, they do not specify the cell line used for this procedure. Was it HEK-T or HEK-FT cells? This information is critical for replicating the methodology and interpreting the results accurately. The authors should clarify the cell line used for the lentiviral packaging.
The CellTiter 96 AQ assay does not measure cell viability directly but is primarily used to analyze cell proliferation. For instance, a cytostatic effect could result in a decrease in absorbance with the CellTiter 96 AQ assay; however, this reduction would not necessarily indicate cell death. Instead, it would reflect a halt in cell proliferation. It is important for authors to differentiate between proliferation and viability when interpreting and reporting results obtained with this assay
This phrase is unclear: "After 12-day culture, the colonies were fixed with methanol and fixed with one after 4-day treatment."
The authors mention that SLC22A3 has been reported to be related to drug efficacy, but they do not specify which drugs or the underlying mechanisms. This is a critical omission, as not all drugs enhance their efficacy in the same way or through the same mechanisms. Providing this information would offer a more comprehensive understanding of the role of SLC22A3 in drug response and its potential as a therapeutic target.
In Figure 1, it is observed that the knockout (KO) of SLC22A3 does not lead to changes in cell proliferation, and the number of colonies is similar to the control. This finding appears contradictory to the statement in the introduction, where it is mentioned that a decrease in this protein correlates with the progression of prostate cancer (PCa) and castration-resistant prostate cancer (CRPC).
The authors used JQ1 (2 µM) and Enzalutamide (2 µM) . But, BRD4 inhibitors are generally not drugs that appear in clinical guidelines for the treatment of prostate cancer (CaP) or castration-resistant prostate cancer (CRPC). Even more importantly, the doses of enzalutamide used are very low.other investigations report that the IC50 of enzalutamide in C4-2 cells is approximately 25 µM. 10.1016/j.tranon.2022.101495.
It would be appropriate to include the dose-response curves for both ENZ and JQ1
Is it possible to discuss homozygosity or heterozygosity in cancer cells that exhibit alterations in chromosome number, as well as chromosomal insertions and deletions? For example, the LNCaP cell line, which is the precursor of C4-2, displays such genomic abnormalities. 10.1159/000015432 In this study, it is observed that, for example, chromosome 6 in LNCaP cells (the precursor cells of C4-2) exhibits segmental insertions and duplications. Since SLC22A3 is located on this chromosome, are there any alterations in the copy number of the SLC22A3 gene?
Author Response
The research is highly relevant to one of the cancers with the highest incidence and mortality rates, particularly regarding its progression to CRPC. The manuscript is clear and concise. However, I consider it important to clarify and improve the following points:
Reply: we really appreciate the reviewer’s positive and thoughtful comments. In this revision, we have modified these accordingly as below.
- In the introduction, the authors state: "This SNP is located on chromosome 6, specifically within the gene SLC22A3, which encodes the organic cation transporter 3 (OCT3), involved in drug metabolism and transportation." Perhaps the authors could delve deeper into the mechanisms underlying the function of SLC22A3 to provide a clearer understanding of its role.
Reply: that is a brilliant point. We have added “Mechanistically, SLC22A3 acts as a passive membrane transporter to facilitate drug diffusion [20].”.
- Similarly, the authors mention that SLC22A3 is increasingly being recognized as an important modulator of human disease and drug response; however, they do not explain the underlying mechanisms by which it exerts these effects.
Reply: that is a great suggestion. We have added “due to that it is involved in drug absorption and elimination”.
- The authors state: "This study highlights the importance of SNP rs9364554 in modulating FOXA1 binding affinity and its pioneering activity, which leads to differential AR binding and transcriptional outcomes, influencing drug efficacy. It provides scientific clues regarding how CRPC or NEPC develop drug resistance through downregulating SLC22A3 transcription. More importantly, the status of SNP rs9364554 can potentially be used as a predictor of drug efficacy for prostate cancer treatment." However, it is important to note that CRPC (castration-resistant prostate cancer) can be approached through various therapeutic strategies, including the use of taxanes, enzalutamide, abiraterone, Radium-233, and sipuleucel-T. The authors may want to address these established treatment modalities to contextualize their findings within the broader therapeutic landscape of CRPC.
Reply: thanks for Reviewer’s thoughtful comments. We have described these established treatment modalities in the manuscript.
- The authors state: "Lentiviral sgRNA constructs as well as packaging vectors were transfected with Lipofectamine 2000 (Thermo Fisher Scientific, Cat# 11668500)." However, they do not specify the cell line used for this procedure. Was it HEK-T or HEK-FT cells? This information is critical for replicating the methodology and interpreting the results accurately. The authors should clarify the cell line used for the lentiviral packaging.
Reply: thanks for Reviewer’s comments. We have added HEK293T cell information accordingly in the section of “Materials and Methods”
- The CellTiter 96 AQ assay does not measure cell viability directly but is primarily used to analyze cell proliferation. For instance, a cytostatic effect could result in a decrease in absorbance with the CellTiter 96 AQ assay; however, this reduction would not necessarily indicate cell death. Instead, it would reflect a halt in cell proliferation. It is important for authors to differentiate between proliferation and viability when interpreting and reporting results obtained with this assay
Reply: thanks for Reviewer’s comments. We have modified the description for the CellTiter 96 AQ assay.
- This phrase is unclear: "After 12-day culture, the colonies were fixed with methanol and fixed with one after 4-day treatment."
Reply: We apologize for this typo issue, which has been fixed in the revision.
- The authors mention that SLC22A3 has been reported to be related to drug efficacy, but they do not specify which drugs or the underlying mechanisms. This is a critical omission, as not all drugs enhance their efficacy in the same way or through the same mechanisms. Providing this information would offer a more comprehensive understanding of the role of SLC22A3 in drug response and its potential as a therapeutic target.
Reply: This is an excellent point. We agree that not all drugs are absorbed into cells in a same way. In this manuscript, we have only tested JQ1 and Enzalutamide; we will test more drugs to see whether SLC22A3 has a preference for certain types of drugs.
- In Figure 1, it is observed that the knockout (KO) of SLC22A3 does not lead to changes in cell proliferation, and the number of colonies is similar to the control. This finding appears contradictory to the statement in the introduction, where it is mentioned that a decrease in this protein correlates with the progression of prostate cancer (PCa) and castration-resistant prostate cancer (CRPC).
Reply: Thanks for pointing out this exceptional observation. The knockout of SLC22A3 alone does not affect the cell proliferation or survival. For prostate cancer and CRPC patients who are more likely to undergo some chemotherapies, the protein level of SLC22A3 is especially important for them. Therefore, we added the following statement in the revision:
“It is noteworthy that SLC22A3 knockout alone does not affect prostate cancer cell proliferation and survival (Figure 1 F-J), suggesting that SLC22A3 is more likely to affect prostate cancer progression via its effects on drug efficacy”.
- The authors used JQ1 (2 µM) and Enzalutamide (2 µM). But BRD4 inhibitors are generally not drugs that appear in clinical guidelines for the treatment of prostate cancer (CaP) or castration-resistant prostate cancer (CRPC). Even more importantly, the doses of enzalutamide used are very low.other investigations report that the IC50 of enzalutamide in C4-2 cells is approximately 25 µM. 10.1016/j.tranon.2022.101495.It would be appropriate to include the dose-response curves for both ENZ and JQ1
Reply: thanks for Reviewer’s thoughtful comments. We agree that BRD4 inhibitors are generally not drugs that appear in clinical guidelines for the treatment of prostate cancer (CaP) or castration-resistant prostate cancer (CRPC). The reason that we chose JQ1 is due to that two of our lab’s previous publications have demonstrated that JQ1 is effective on inhibiting prostate cancer growth. In this study, we used this drug as a tool to prove the principle.
Zhang, P., Wang, D., Zhao, Y. et al. Intrinsic BET inhibitor resistance in SPOP-mutated prostate cancer is mediated by BET protein stabilization and AKT–mTORC1 activation. Nat Med 23, 1055–1062 (2017).
Yan Y, Ma J, Wang D, Lin D, Pang X, Wang S, Zhao Y, Shi L, Xue H, Pan Y, Zhang J, Wahlestedt C, Giles FJ, Chen Y, Gleave ME, Collins CC, Ye D, Wang Y, Huang H. The novel BET-CBP/p300 dual inhibitor NEO2734 is active in SPOP mutant and wild-type prostate cancer. EMBO Mol Med. 2019 Nov 7;11(11):e10659.
Regarding the does-response curves, we agree that does-response curves would be an ideal method to demonstrate the IC50 for drug treatment. However, the data from our current experiments (three concentrations) are not enough to generate dose-response curves.
- Is it possible to discuss homozygosity or heterozygosity in cancer cells that exhibit alterations in chromosome number, as well as chromosomal insertions and deletions? For example, the LNCaP cell line, which is the precursor of C4-2, displays such genomic abnormalities. 10.1159/000015432In this study, it is observed that, for example, chromosome 6 in LNCaP cells (the precursor cells of C4-2) exhibits segmental insertions and duplications. Since SLC22A3 is located on this chromosome, are there any alterations in the copy number of the SLC22A3 gene?
Reply: Thank you for Reviewer’s insightful question. Indeed, prostate cancer cells often exhibit alterations in chromosome number, as well as chromosomal insertions and deletions, which can impact gene expression and function. Specifically, regarding SLC22A3 copy number variants, we observed that the ratio of C allele/T allele in our Sanger sequencing results (Figure 2C) is different from 22Rv1 and VCaP, although four of these cell lines are considered as SLC22A3 heterozygous. This result might align with the Reviewer’s question that SLC22A3 might be also influenced by copy number variants. However, we acknowledge that further studies are needed to fully understand whether this difference is affected by copy number difference.
Reviewer 2 Report
Comments and Suggestions for Authors
In this manuscript, the authors emphasize the potential use of this SNP as a predictor of drug efficacy in prostate cancer patients. They investigated the correlation between SNP rs9364554, androgen receptor binding, and drug response. The study concept is strong, and the introduction is well-articulated. The methodology is clear; however, the dataset statistics are not included. The references are recent and appropriately integrated into the text. Some issues that require clarification are listed below:
1. In the introduction section
- In line number 36, “AR signaling playing a pivotal role in…” AR full name must be written before being used as an abbreviation.
- Also, “GWAS studies” the abbreviation must be defined at its first appearance in the text.
2. In the results section
- The labels on the lanes and the figure legends were missing in the supplementary figures (original films for WB and EMSA).
- The sample size (n) for both normal and tumor samples is not clearly detailed in the methods section. This creates confusion regarding whether these are human samples or cell lines. To enhance clarity, the authors should distinguish between the statistical analyses for the datasets and those for the cell lines.
- In Figure 1 (F-j), what do #1 and #2 refer to? Additionally, please note that SLC22A3 should be written with a capital "A" rather than a lowercase "a."
- In Figure 2E, the authors should clarify their reasoning for considering the lower band as a mutant band for the FOXA1 protein in DU145.
- The supplementary figures should be inserted into the text in suitable places for a clearer explanation.
Author Response
Comments and Suggestions for Authors
In this manuscript, the authors emphasize the potential use of this SNP as a predictor of drug efficacy in prostate cancer patients. They investigated the correlation between SNP rs9364554, androgen receptor binding, and drug response. The study concept is strong, and the introduction is well-articulated. The methodology is clear; however, the dataset statistics are not included. The references are recent and appropriately integrated into the text. Some issues that require clarification are listed below:
Reply: we really appreciate the reviewer’s positive and thoughtful comments. In this revision, we have modified these accordingly.
- In the introduction section
- In line number 36, “AR signaling playing a pivotal role in…” AR full name must be written before being used as an abbreviation.
- Also, “GWAS studies” the abbreviation must be defined at its first appearance in the text.
Reply: thanks for pointing this out. We have spelled these out in the revision.
- In the results section
- The labels on the lanes and the figure legends were missing in the supplementary figures (original films for WB and EMSA).
Reply: thanks for this point. We have added figure legends for this supplementary figure ((original films for WB and EMSA).
- The sample size (n) for both normal and tumor samples is not clearly detailed in the methods section. This creates confusion regarding whether these are human samples or cell lines. To enhance clarity, the authors should distinguish between the statistical analyses for the datasets and those for the cell lines.
Reply: thanks for this great point. We have added detailed information in the figure legends.
- In Figure 1 (F-J), what do #1 and #2 refer to? Additionally, please note that SLC22A3 should be written with a capital "A" rather than a lowercase "a."
Reply: thanks for this excellent point. The #1 and #2 are two constructs for sgSLC22A3. We have described this in the figure legend in the revision. Also, the “SLC22a3” has been corrected to “SLC22A3”.
- In Figure 2E, the authors should clarify their reasoning for considering the lower band as a mutant band for the FOXA1 protein in DU145.
Reply: thanks for this thoughtful point. We suspected that this lower band is a mutant FOXA1 band due to two reasons: 1) our specifically recognized FOXA1 antibody detect this band; 2) FOXA1 is highly mutated in prostate cancer.
- The supplementary figures should be inserted into the text in suitable places for a clearer explanation.
Reply: thanks for this great suggestion. We have inserted all supplementary figures into the supplementary file.